# Mono-methylated histones control PARP-1 in chromatin and transcription

**Gbolahan Bamgbose[1], Guillaume Bordet[1], Niraj Lodhi[2], Alexei Tulin[1]***

[1]Department of Biomedical Sciences, School of Medicine and Health Sciences, University of North Dakota, Grand Forks, United States; [2]Fox Chase Cancer Center, Philadelphia, United States

**\*For correspondence:**
alexei.tulin@und.edu

**Competing interest:** The authors declare that no competing interests exist.

**Abstract** PARP-1 is central to transcriptional regulation under both normal and stress conditions, with the governing mechanisms yet to be fully understood. Our biochemical and ChIP-seq-based analyses showed that PARP-1 binds specifically to active histone marks, particularly H4K20me1. We found that H4K20me1 plays a critical role in facilitating PARP-1 binding and the regulation of PARP-1-dependent loci during both development and heat shock stress. Here, we report that the sole H4K20 mono-methylase, *pr-set7*, and *parp-1 Drosophila* mutants undergo developmental arrest. RNA-seq analysis showed an absolute correlation between PR-SET7- and PARP-1-dependent loci expression, confirming co-regulation during developmental phases. PARP-1 and PR-SET7 are both essential for activating *hsp70* and other heat shock genes during heat stress, with a notable increase of H4K20me1 at their gene body. Mutating *pr-set7* disrupts monomethylation of H4K20 along heat shock loci and abolish PARP-1 binding there. These data strongly suggest that H4 monomethylation is a key triggering point in PARP-1 dependent processes in chromatin.

## eLife assessment

This **valuable** study presents **convincing** evidence for an association between PARP-1 and H4K20me1 in transcriptional regulation, supported by biochemical and ChIP-seq analyses. The work contributes significantly to our understanding of how Parp1 associates with target genes to regulate their expression.

## Introduction

The chromosomes of eukaryotes are made up of chromatin, which is a complex of DNA tightly wound around nucleosomes (*Kouzarides, 2007*). Nucleosomes are the basic unit of chromatin, and they are made up of about 147 bp of DNA wrapped around an octamer of four core histones (H2A, H2B, H3, and H4) (*Kouzarides, 2007*). Eukaryotic DNA is packed into nucleosomes to maintain higher-order chromatin structure. As a consequence, DNA sequences are inaccessible to a plethora of protein effectors for biological processes such as DNA damage repair, replication, and transcription (*Peterson and Laniel, 2004*). To overcome the nucleosome barrier, eukaryotes have developed several mechanisms that ensure the successful removal of nucleosomes. One of the key effectors involved in chromatin remodeling is Poly(ADP)-ribose Polymerase 1 (PARP-1), a chromatin-associated nuclear enzyme implicated in a host of biological processes including, transcriptional regulation and DNA repair (*Kraus and Hottiger, 2013*; *Posavec Marjanović et al., 2017*; *Thomas and Tulin, 2013*). PARP-1 catalyzes the addition of poly(ADP)-ribose (pADPr) moieties onto acceptor proteins by utilizing donor NAD$^+$ as substrate. Upon PARP-1 activation due to developmental and environmental cues or stress, PARP-1 is auto modified, and it attaches negatively charged poly(ADP)-ribose polymers onto histones and other chromatin-associated proteins, thereby decondensing DNA via electro-repulsion and facilitating

PARP-1 mediated biological processes (*D'Amours et al., 1999*; *Tulin and Spradling, 2003*). Besides PARP-1, histone modifications are essential for remodeling chromatin structure. Histones are subject to post-translational modifications, including phosphorylation, acetylation, sumoylation, ubiquitylation, and ADP-ribosylation on their amino-terminal tails protruding from the nucleosome core (*Peterson and Laniel, 2004*; *Berger, 2002*; *Vaquero et al., 2003*). Modifications of these histone tails can alter chromatin structure by modifying inter-nucleosome interactions or by serving as docking sites for effectors to facilitate diverse biological outcomes depending on the histone(s) so modified (*Peterson and Laniel, 2004*; *Bannister and Kouzarides, 2011*).

The binding of PARP-1 to post-translationally modified histones may be necessary for PARP-1-mediated biological processes and PARP-1 targeting to chromatin. In our previous study, we showed that the phosphorylation of the histone variant H2Av directly impacts the localization, activation, and enzymatic activity of PARP-1 (*Kotova et al., 2011*). Also, we previously showed that the N-terminal tails of histones H3 and H4 are needed for the enzymatic activation of PARP-1 (*Pinnola et al., 2007*). Taken together, there is strong evidence that histone modifications can regulate PARP-1 activity in vivo.

In this study, we demonstrate that PARP-1's transcriptional regulatory role under regular and heat stress conditions may be intricately tied to its interaction with specific mono-methylated active histone marks, notably H4K20me1, H3K4me1, H3K36me1, H3K27me1, and H3K9me1 that it is inhibited by repressive H3K9me2/3 marks. Through transcriptomic analysis of *pr-set7* and *parp-1* mutant *Drosophila* third-instar larvae, we uncover a high correlation in differentially expressed genes and co-enrichment of PARP-1 and H4K20me1 at a subset of these genes, underscoring the role of PR-SET7/H4K20me1 in facilitating PARP-1-mediated transcriptional regulation. Importantly, we show that PARP-1 and PR-SET7 are essential for the activation of specific heat shock genes including *hsp70*, and the dynamic regulation of H4K20me1 during this process, further validates our hypothesis of H4K20me1 being crucial for PARP-1 binding and gene regulation under developmental and heat stress conditions.

## Results

### PARP-1 binds to H4K20me1, H3K4me1, H3K36me1, H3K9me1, and H3K27me1 in vitro and in vivo

In this study, we investigated the interaction between full-length recombinant PARP-1 and various histone modifications using a histone peptide array containing modified histone peptides individually and in combination with other peptides (*Figure 1A*). Our previous work established that PARP-1 binds to core histones (*Pinnola et al., 2007*). Here, we aimed to determine the specific histone marks that modulate PARP-1's affinity for chromatin. We found that PARP-1 binds specifically to spots containing H4K20me1, H3K4me1, H3K36me1, H3K9me1, or H3K27me1 peptides (*Figure 1B, C and D* and *Supplementary file 1*). Specificity analysis showed that PARP-1 binding was highly specific for H3K4me1, H3K36me1, H3K9me1, and H3K27me1 containing peptides with the highest specificity observed for H4K20me1 (*Figure 1E* and *Supplementary file 1*). Additionally, we explored the inhibitory effects of different histone modifications on PARP-1 binding. Our findings showed that H3K9me2/3 hindered PARP-1 binding to spots containing H3K4me1 (*Figure 1F*, *Figure 1—figure supplements 1 and 2*). Conversely, H3K4me1 enhanced PARP-1 binding to spots with H3K9me2/3 (*Figure 1F*, *Figure 1—figure supplement 1*). In parallel, our data shows that the phosphorylation of H3S10 or H3T11, two residues nearby H3K9 diminish PARP-1 binding to H3K9me1 (*Figure 1—figure supplements 1 and 2*). Similarly, the phosphorylation of H3K28 affects PARP-1 binding to H3K27me1 (*Figure 1—figure supplements 1 and 2*).

Next, we examined the association of PARP-1 with H4K20me1, H3K4me1, H3K36me1, H3K9me1, H3K9me2/3, and H3K27me1 in *Drosophila* third-instar larvae using ChIP-seq data. PARP-1 peaks were highly correlated with H4K20me1, H3K4me1, H3K36me1, H3K9me1, and H3K27me1 but significantly less correlated with H3K9me2/3 peaks (*Figure 1G*). Consistently, at gene clusters, H4K20me1 and H3K4me1 were highly enriched at gene bodies in cluster I which also had the highest PARP-1 enrichment at promoters. H3K36me1, H3K9me1, and H3K27me1 were also enriched in cluster I albeit to a lesser degree than H4K20me1 and H3K4me1 (*Figure 1H*). In contrast, H3K9me2/3 were more enriched at clusters II and III but depleted in cluster I (*Figure 1H*). Additionally, PARP-1 associates with H4K20me1 and H3K4me1 on *Drosophila* polytene chromosomes (*Figure 2*). Notably, H4K20me1,

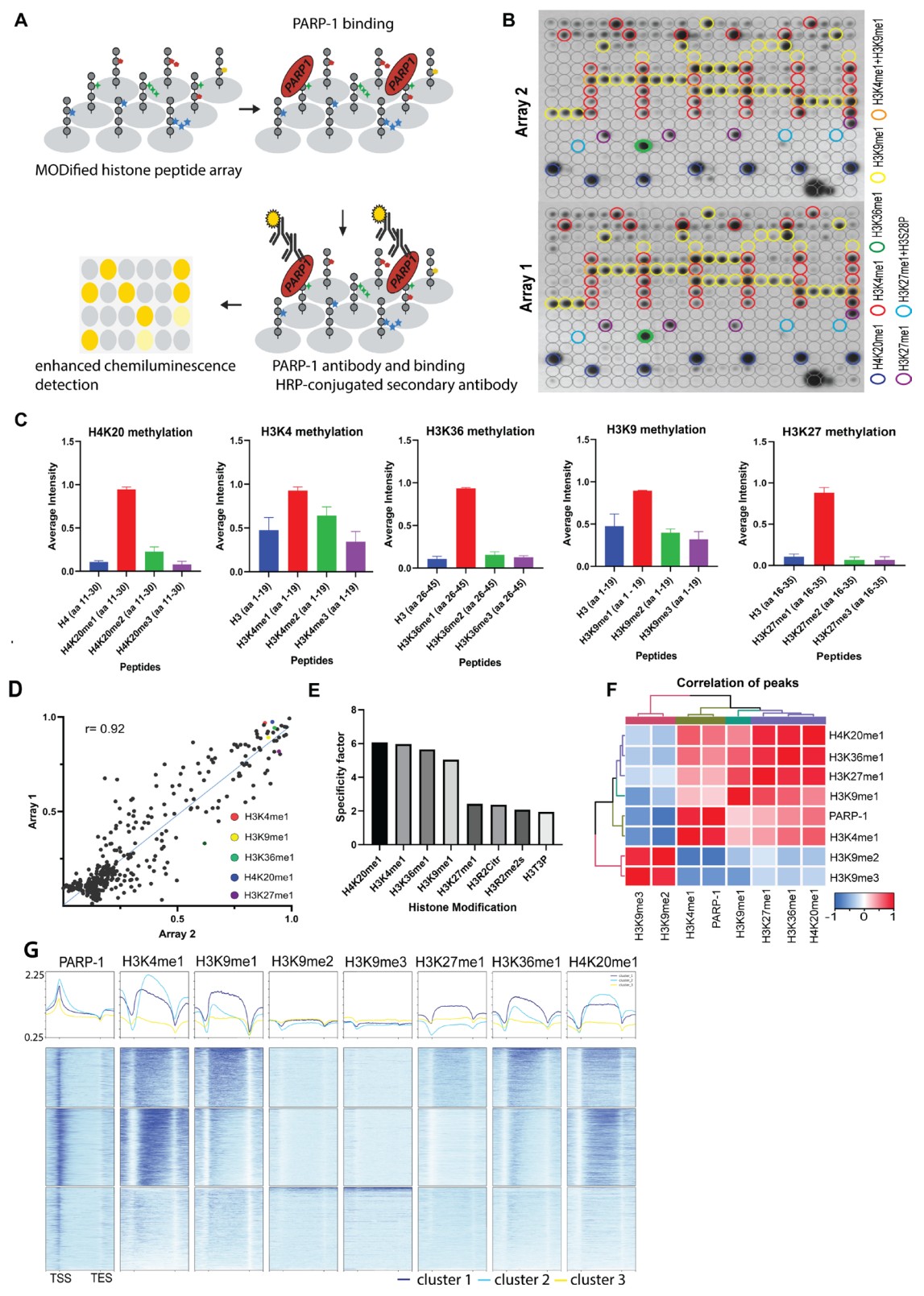

**Figure 1.** PARP-1 binds H4K20me1, H3K4me1, H3K36me1, H3K9me1, and H3K27me1 in vitro and in vivo. (**A**) Illustration of the MODified histone peptide assay (Active Motif) used to determine PARP-1 binding to histone modifications. The histone peptide array (Top, left), comprising 19mer peptides with single or up to four concurrent histone modifications, was employed to investigate PARP-1's binding affinity for histone modifications and to assess the impact of adjacent modified peptides on PARP-1 binding. This array was first blocked, then incubated with PARP-1 protein (Top, right).

*Figure 1 continued on next page*

*Figure 1 continued*

Subsequently, it was stained with a PARP-1 antibody and a horseradish peroxidase (HRP)-conjugated secondary antibody (Bottom, right). Visualization of PARP-1 binding was done through enhanced chemiluminescence detection and captured on X-ray film (Bottom, left). See Methods for a full description. (**B**) Signal intensity on modified histone peptide array based on incubation with PARP-1 protein. (**C**) Average intensities of PARP-1 binding to single histone peptides. Data are presented as mean ± SEM (**D–E**) Reproducibility and specificity of spot intensities from modified histone peptide array duplicates. (**D**) Scatter plot showing the correlation of the average intensities of duplicate arrays. Intensities of PARP-1 binding to all peptides and spots containing single H4K20me1, H3K4me1, H3K36me1, H3K9me1, and H3K27me1 (key) are shown. Pearson's correlation coefficient (r) is 0.92. (**E**) Bar chart showing top 8 histone modifications with the highest specificity for PARP-1 binding. The specificity factor was calculated by dividing the average intensity of spots that contain the modified histone peptide by the average intensity of spots that do not contain the peptide. (**F**) Spearman correlation of PARP-1, H4K20me1, H3K4me1, H3K36me1, H3K9me1, H3K9me2/3, and H3K27me1 peaks in *Drosophila* third-instar larvae based on fraction of overlap. (**G**) Heatmaps showing k-means clustering-generated occupancy of PARP-1, H4K20me1, H3K4me1, H3K36me1, H3K9me1, H3K9me2/3, and H3K27me1 normalized ChIP-seq signals in third-instar larvae at *Drosophila* genes. ChIP-seq signals are sorted in descending order. The upper plots show the summary of the signals.

The online version of this article includes the following figure supplement(s) for figure 1:

**Figure supplement 1.** PARP-1 binding is inhibited by H3K9me2/3 peptides or by phosphorylation of adjacent residues.

**Figure supplement 2.** Inhibition of PARP-1 binding by H3K9me2/3 and phosphorylation of adjacent residues.

H3K4me1, H3K36me1, H3K9me1, and H3K27me1 are associated with active genes while H3K9me2/3 is associated with low-expression or silent genes (*Barski et al., 2007*). Thus, our data shows that PARP-1 predominantly binds to active mono-methylated histone marks, specifically H4K20me1, H3K4me1, H3K36me1, H3K9me1, and H3K27me1 while also suggesting that the binding of PARP-1 may be hindered by repressive histone marks such as H3K9me2/3.

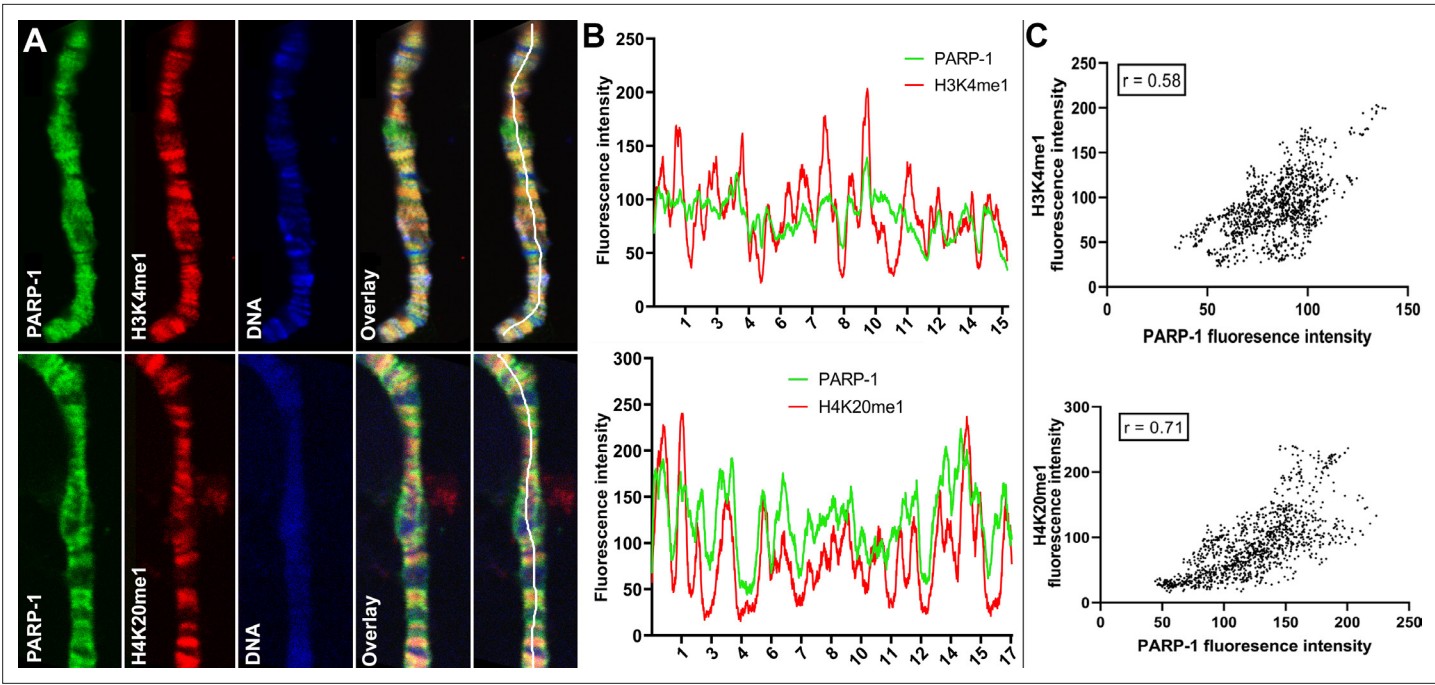

**Figure 2.** PARP-1 colocalizes with H3K4me1 and H4K20me1 histone marks in polytene chromosomes. (**A**) Immunofluorescent staining of *Drosophila* salivary gland chromosomes showing colocalization of PARP-1 with H3K4me1 and H4K20me1. White lines indicate areas of the colocalization-quantification shown in (**B**). (**B–C**) Quantification of fluorescence intensity of PARP-1, H3K4me1, and H4K20me1 at *Drosophila* polytene chromosomes in panel A. (**B**) Images show the distribution of PARP-1 fluorescence Intensity with H3K4me1 (top) and H4K20me1 (bottom) fluorescence intensity. (**C**) Images represent a scatterplot showing PARP-1 fluorescence intensity with H3K4me1 (top) and H4K20me1 (bottom) fluorescence intensity. Pearson correlation coefficients (**r**) are respectively 0.58 (top) and 0.71 (bottom).

## PARP-1 and PR-SET7 co-regulate developmental gene expression programs

Next, we examined the relationship between H4K20me1 and PARP-1 during transcriptional regulation. Given that flies that lack PR-SET7, the sole methylase of H4K20, and PARP-1 undergo developmental arrest during the larval-to-pupal transition (*Karachentsev et al., 2005*; *Kotova et al., 2010*), we hypothesized that PR-SET7/H4K20me1 and PARP-1 may regulate similar gene expression programs during development. To validate our hypothesis, we initially confirmed that the *pr-set7[20]* mutant not only eliminated PR-SET7 RNA and protein but also abrogated H4K20me1 modification (*Figure 3—figure supplement 1*). Interestingly, in the absence of PARP-1, neither PR-SET7 RNA nor protein levels were affected (*Figure 3—figure supplements 2 and 3*), indicating that PARP-1 is not directly implicated in the regulation of *pr-set7*. This finding contrasts with recent evidence demonstrating PARP1-induced degradation of PR-SET7/SET8 in human cells (*Estève et al., 2022*). Subsequently, we integrated PARP-1 and H4K20me1 ChIP-seq data in third-instar larvae and RNA-seq of *pr-set7[20]* and *parp-1[C03256]* mutant third-instar larvae. Our data showed that the differentially expressed genes (DEGs) in *pr-set7[20]* and *parp-1[C03256]* transcriptome were highly correlated (Pearson, $r=0.79$) (*Figure 3A and B*). However, despite the global reduction of H4K20me1, and PARP-1 RNA in *pr-set7[20]* and *parp-1[C03256]* mutants, respectively (*Karachentsev et al., 2005*; *Kotova et al., 2010*), there was a limited alteration in gene expression at genes where PARP-1 was co-localized with H4K20me1 (*Figure 3C and D*). Differentially expressed genes in both *pr-set7[20]* and *parp-1[C03256]* mutants and co-enriched with PARP-1 and H4K20me1 were mainly upregulated (n=101, 72%) (*Figure 3E*). Intriguingly, under wild-type conditions, these genes displayed expression levels approximately 40% higher than the average and demonstrated increased RNA-Polymerase II occupancy both at their promoter regions and gene bodies compared to other genes (*Figure 3—figure supplement 4*), indicating their high activity in wild-type context. Interestingly, among the genes co-enriched with PARP-1 and H4K20me1, those only differentially expressed in *parp-1[C03256]* mutants were mostly upregulated (n=94, 85%), while those only differentially expressed in *pr-set7[20]* mutants were mainly downregulated (n=191, 67%), indicating that PARP-1 and H4K20me1 also have distinct functions in gene regulation at co-enriched genes (*Figure 3E*). Gene ontology analysis of differentially expressed genes bound by PARP-1 and H4K20me1 showed that metabolic genes were upregulated while genes involved in neuron development and morphogenesis were downregulated in both *pr-set7[20]* and *parp-1[C03256]* mutants (*Figure 3F*). Finally, to extend the generalizability of our observations beyond *Drosophila*, we compared the distribution of PARP1 and H4K20me1 in Human K562 cells. Strikingly, we observed a correlation in their distribution, suggesting that the interplay between PARP-1 and H4K20me1 is not limited to fruit flies (*Figure 3—figure supplement 5*). Overall, our results indicate H4K20me1 may be required for PARP-1 binding to preferentially repress metabolic genes and activate genes involved in neuron development at co-enriched genes.

## PARP-1 and PR-SET7/H4K20me1 are required for the optimal expression of heat shock genes during heat stress

Next, we examined PARP-1 and H4K20me1 controlled gene expression programs during dynamic transcriptional changes. During heat shock, PARP-1 spreading from the promoter to the gene bodies of *hsp70* facilitates nucleosomal loss which leads to transcriptional activation (*Petesch and Lis, 2012*; *Thomas et al., 2019*). Since H4K20me1 is mostly enriched in gene bodies (*Figure 1H*), we hypothesized that H4K20me1 may be required for PARP-1 spread at *hsp70* during heat shock. First, we examined the expression of heat shock genes before and after 30 min of heat shock in WT, *parp-1[C03256]*, and *pr-set7[20]* third-instar larvae. The expression of heat shock genes (*hsp22, hsp23, hsp68, hsp83*) including *hsp70* were significantly reduced in *parp-1[C03256]* and *pr-set7[20]* animals after heat shock (*Figure 4A*). ChIP-seq results revealed that PARP-1 highly occupied the promoters of *hsp23, hsp68, hsp70*, and *hsp83*, as well as in the gene body of *hsp22*, prior to heat shock (*Figure 4B and C*). Analysis of H4K20me1 enrichment showed low levels across the gene bodies of *hsp22, hsp68*, and *hsp70* (group A) but high levels at gene bodies of *hsp23* and *hsp83* (group B) before heat shock (*Figure 4B and C*). Based on these findings, we hypothesized that an increase in H4K20me1 enrichment in the gene bodies of group A genes would facilitate PARP-1 binding after heat shock, while H4K20me1 enrichment remains unchanged in group B genes thereby facilitating PARP-1 binding and spreading. To our surprise, H4K20me1 levels increased at the gene bodies of group A genes (low H4K20me1

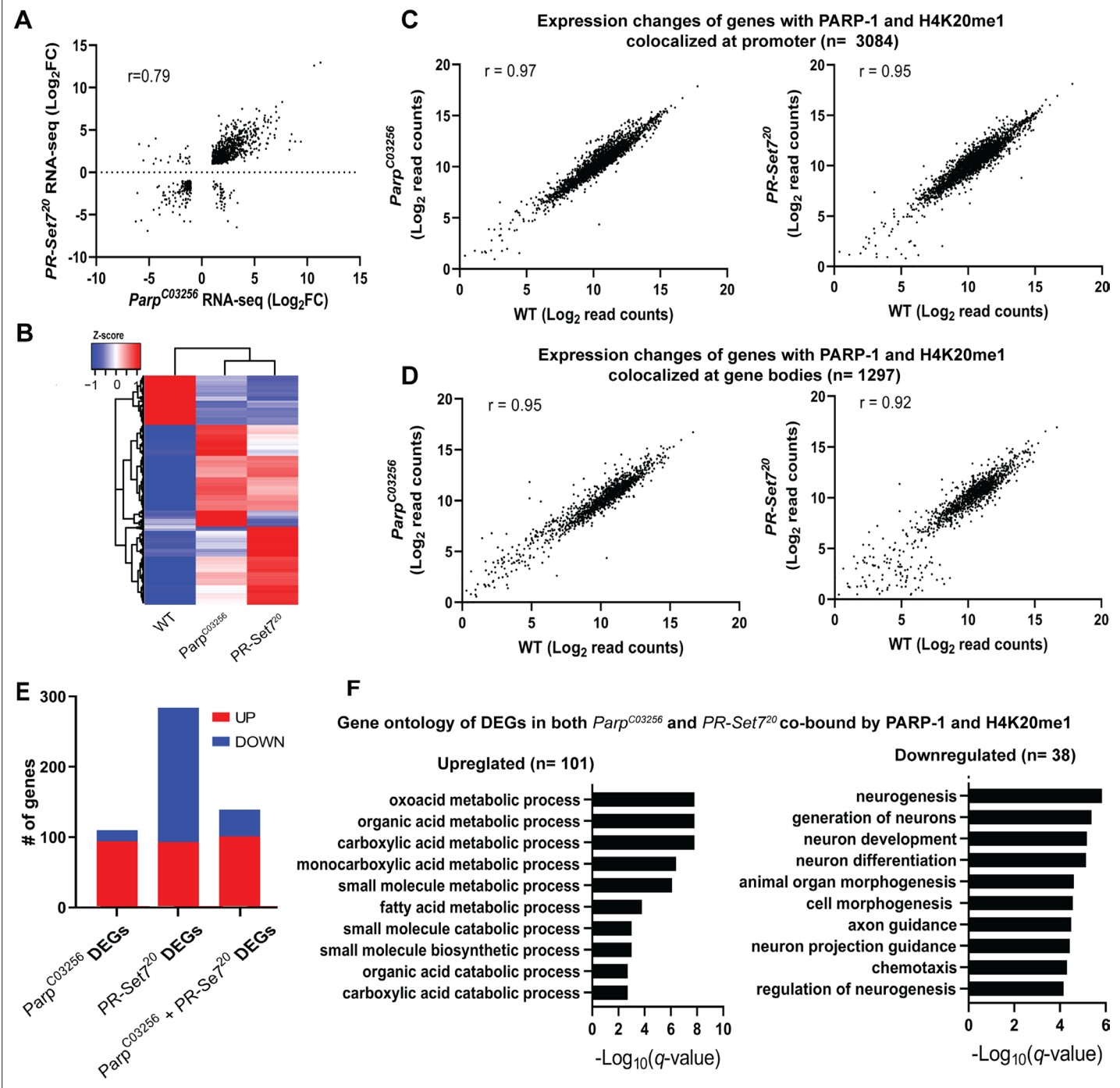

**Figure 3.** PARP-1 and H4K20me1 are required for the repression of metabolic genes and activation of developmental genes at co-enriched genes. (**A**) Scatterplot plot showing correlation of differentially expressed genes (DEGs) in *parp-1[C03256]* and *pr-set7[20]* (Pearson's *r*=0.79). (**B**) Heatmap showing the normalized read counts of DEGs in both *parp-1[C03256]* and *pr-set7[20]*. Normalized read counts are shown as row z-scores. (**C–D**) Dot plots showing transcriptional changes of genes co-enriched with PARP-1 and H4K20me1 in *parp-1[C03256]* and *pr-set7[20]* compared to WT at promoters (**C**) and gene bodies (**D**). (**E**) Summary of DEGs in *parp-1[C03256]* and *pr-set7[20]* and both mutants that were co-enriched with PARP-1 and H4K20me1. (**F**) Gene ontology of upregulated (left) and downregulated (right) DEGs in both *parp-1[C03256]* and *pr-set7[20]* mutants that were co-enriched with PARP-1 and H4K20me1.

The online version of this article includes the following source data and figure supplement(s) for figure 3:

**Figure supplement 1.** Validation of pr-set720 mutant.

**Figure supplement 1—source data 1.** Original file for the Western blot analysis in *Figure 3—figure supplement 1*.

**Figure supplement 1—source data 2.** *Figure 3—figure supplement 1* and original scans of the relevant Western blot analysis with highlighted bands

*Figure 3 continued*

and sample labels.

**Figure supplement 2.** PR-SET7 expression level is not affected in *parp-1^C03256* mutant.

**Figure supplement 3.** PR-SET7 protein level is not affected in *parp-1^C03256* mutant.

**Figure supplement 3—source data 1.** Original file for the Western blot analysis in *Figure 3—figure supplement 3*.

**Figure supplement 3—source data 2.** This file containing *Figure 3—figure supplement 3* and original scans of the relevant Western blot analysis with highlighted bands and sample labels.

**Figure supplement 4.** Genes coenriched in H4K20me1 and PARP-1 that are upregulated in both *parp-1* and *pr-set7* mutants are highly expressed genes.

**Figure supplement 5.** PARP-1 binding correlates with H4K20me1 enrichment in Human K562 cells.

before heat shock) (*Figure 4D*) but decreased significantly at group B genes (high H4K20me1 before heat shock) after heat shock (*Figure 4E*). These findings suggest that dynamic changes in H4K20me1 enrichment may regulate the expression of heat shock genes during heat shock and indicate that H4K20me1 may not be necessary for PARP-1 binding and PARP-1-mediated activation of group B genes during heat shock.

## Mutating PR-SET7 methyl transferase disrupts PARP-1 binding to chromatin and diminishes poly(ADP)-ribosylation levels during heat shock stress

Our results showed that prior to heat shock, there was no effect on PARP-1 binding in both WT and *pr-set7^20* background animals to chromatin (*Figure 5A*). However, after heat shock, we observed that PARP-1 binding was concentrated at specific loci in WT animals, while it was diminished from chromatin in *pr-set7^20* animals (*Figure 5A*). These findings suggest that PR-SET7/H4K20me1 plays a crucial role in mediating PARP1 binding to chromatin during heat stress, and thus, controls PARP1-dependent processes in chromatin. Previous studies showed that heat shock results in pADPr accumulation at the heat shock-induced puffs during PARP-1-dependent transcriptional activation (*Thomas and Tulin, 2013*; *Tulin and Spradling, 2003*; *Kotova et al., 2011*; *Petesch and Lis, 2012*; *Thomas et al., 2019*; *Petesch and Lis, 2008*). Therefore, we tested if heat-shock treatment alters pADPr accumulation in *pr-set7^20* animals. As expected, the pADPr level increased by 26.1 times after heat-shock treatment in wild-type animals, while in *pr-set7^20* mutants the level of pADPr increased only 2.3 folds (*Figure 5B*). This observation implies that PR-SET7/H4K20me1 plays a key role in regulating PARP-1 upon environmental stresses such as heat shock.

## Discussion

This study illuminates the intricate interactions between PARP-1 and diverse histone modifications, with a particular emphasis on the influence of histone marks on the chromatin binding propensity of PARP-1. We discovered an elevated binding affinity of PARP-1 towards specific mono-methylated active histone modifications such as H4K20me1, H3K4me1, H3K36me1, H3K9me1, and H3K27me1 in vitro (*Figure 1B and C*). Intriguingly, our results also suggest that the repressive histone modifications, H3K9me2/3 may potentially hinder PARP-1 binding to chromatin (*Figure 1F*). The association between PARP-1 peaks and distinct histone marks in *Drosophila* third-instar larvae, as revealed through ChIP-seq data, corroborates the significance of these histone marks in PARP-1 binding. This is predominantly apparent for H4K20me1, H3K4me1, H3K36me1, H3K9me1, and H3K27me1, while H3K9me2/3 demonstrated a lesser correlation (*Figure 1G*), further validating the repressive role of these marks.

We additionally examined the potential functional consequences of PARP-1's interaction with these histone marks, particularly H4K20me1. Mono-methylation of histone H4 on lysine 20 (H4K20me1), catalyzed by PR-SET7, is evolutionarily conserved from yeast to humans (*Oda et al., 2010*; *Wang and Jia, 2009*). H4K20me1, similar to PARP-1, is enriched at highly active genes (*De Angelis Campos et al., 2011*; *Gibson et al., 2016*; *Krishnakumar and Kraus, 2010*; *Nalabothula et al., 2015*; *Nikolaou et al., 2017*; *Talasz et al., 2005*; *Vakoc et al., 2006*). Also, H4K20me1 is critical for transcriptional regulation, maintenance of genome integrity, cell cycle regulation, and double-strand break

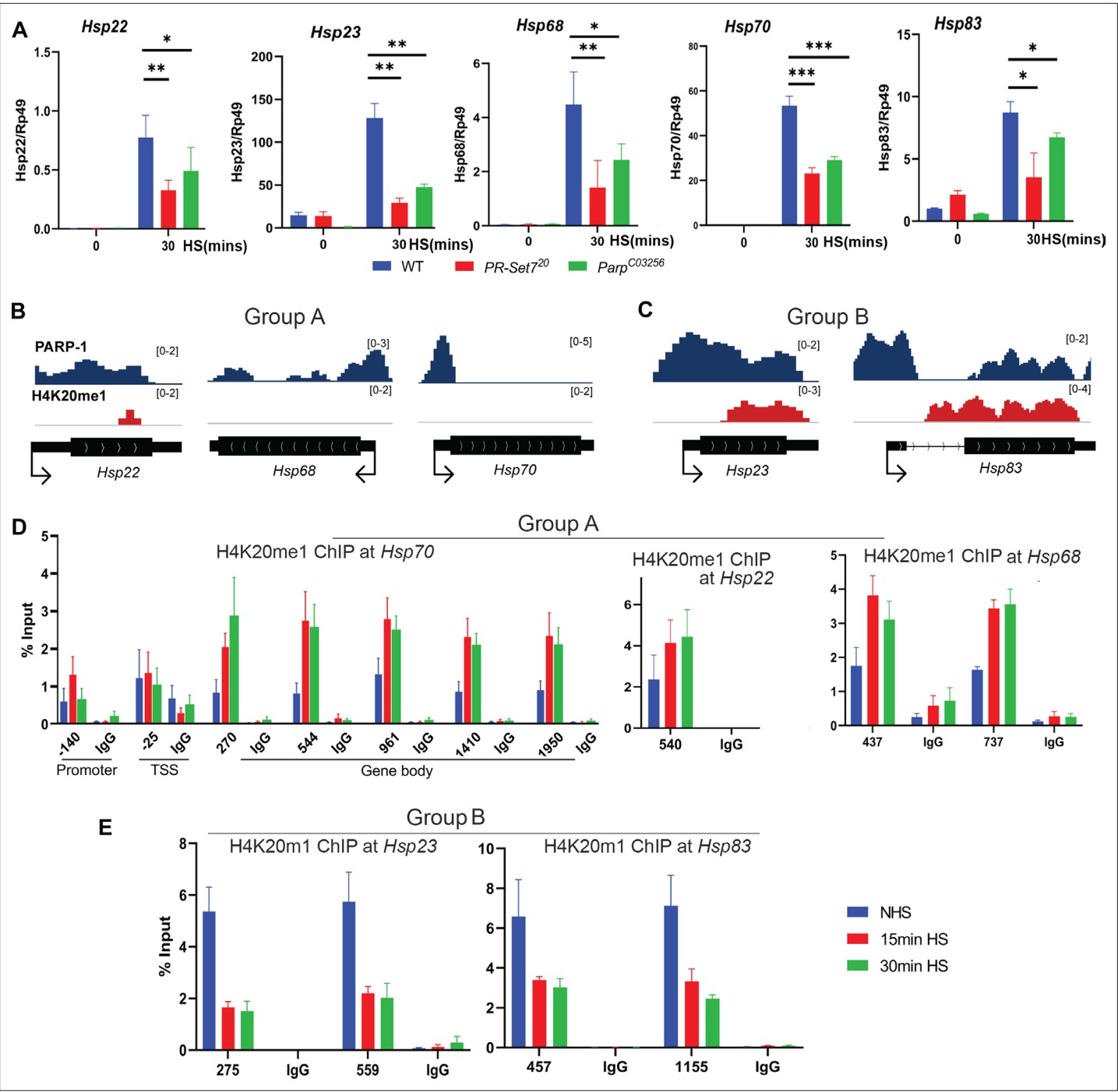

**Figure 4.** Dynamic H4K20me1 enrichment regulates the expression of heat-shock genes during heat shock. (**A**) Expression of *hsp22*, *hsp23*, *hsp68*, *hsp70*, and *hsp83* in wild-type (WT), *parp-1^C03256^* and *pr-set7^20^* third-instar larvae before and after 30 min heat shock. Data shown are from three to five biological replicates. \*\*\*p<0.001, \*\*p<0.01, \*p<0.05 (Unpaired t-test; Two-tailed). Integrative Genome Viewer (IGV) tracks showing normalized ChIP-seq tracks of PARP-1 and H4K20me1 in third-instar larvae at (**B**) *hsp22, hsp68, hsp68, hsp70,* and (**C**) *hsp22, hsp83* before heat shock. Mononucleosome ChIP-qPCR in WT showing enrichment of H4K20me1 at (**D**) *hsp70, hsp22, hsp68,* and (**E**) *hsp23* and *hsp83* before heat shock (NHS), after 15 min heat shock and 30 min heat shock. Primers used spanned the *hsp70* locus and the gene bodies of *hsp22, hsp23, hsp68, hsp70,* and *hsp83*. Data shown are from three biological replicates. Data are presented as mean ± SEM.

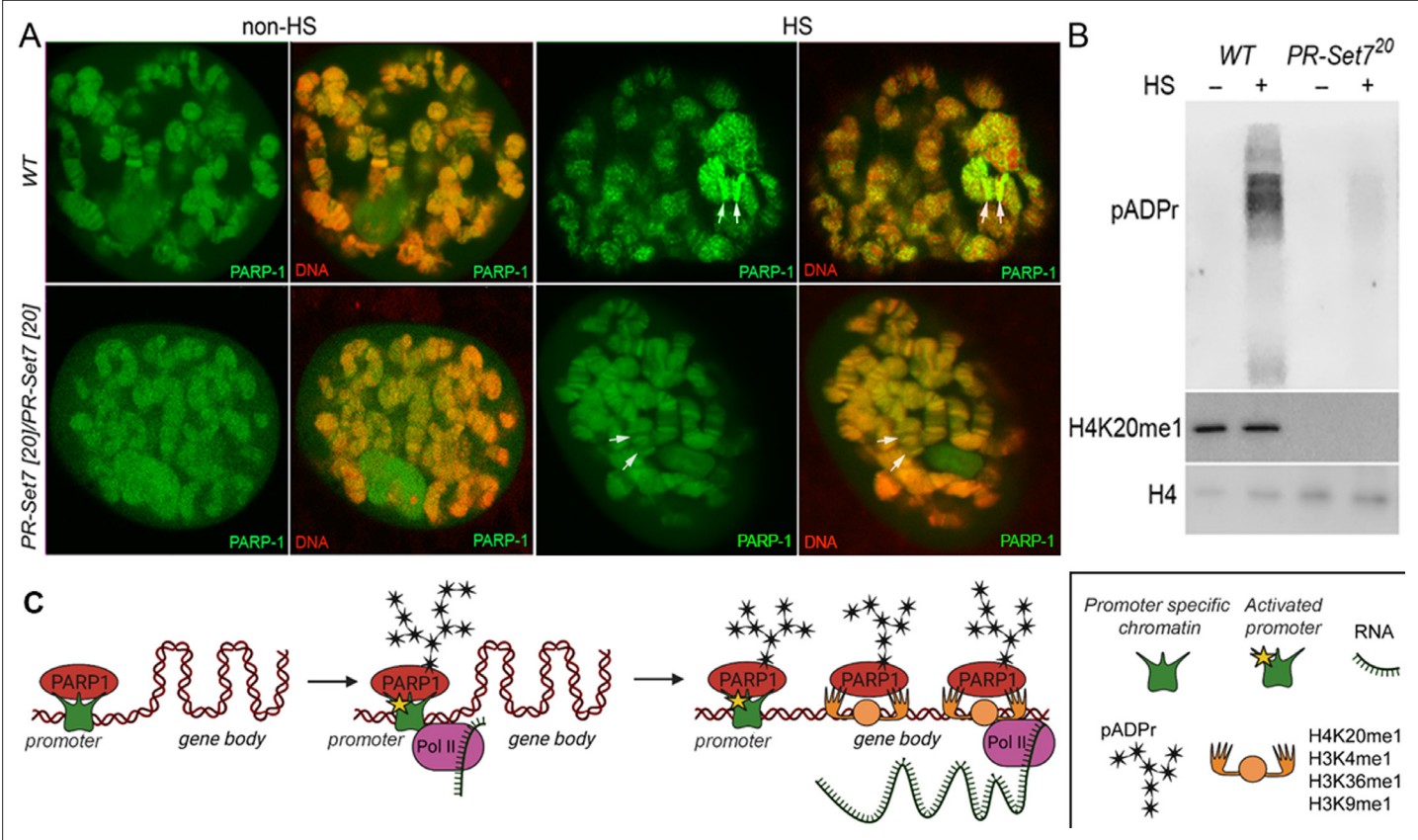

**Figure 5.** Mono-methylated histones controls PARP-1 binding along gene body to regulate transcription. (**A**) PR-SET7/H4K20me1 is required for holding PARP-1 in chromatin during heat shock (HS). PARP-1 (green) protein recruitment to *hsp70* locus (Arrows) in salivary gland polytene nuclei in wild-type (WT) and *pr-set7[20]* mutant third instar larvae. A single salivary gland polytenized nucleus of wandering third instar larvae is shown for each genotype. Wild-type genotype - UASt::PARP-1-EYFP, GAL4[Mz1087.hx]; *pr-set7* mutant genotype – UASt::PARP-1-EYFP, GAL4[Mz1087.hx]; *pr-set7[20]*. (**B**) Equal amounts of lysates from the wild-type expressing PARP-1-EYFP (WT) and *pr-set7[20]* mutants expressing PARP-1-EYFP (*pr-set7[20]*) grown at 22 °C or heat shocked at 37 °C for 1 hr at the third-instar larvae stage were subjected to immunoblot analysis using mouse anti-pADPr (10 H), anti-H4K20me1 and anti-histone H4 (loading control) antibodies. (**C**) Model of PARP-1 regulation by histone modifications. PARP-1 binds to a nucleosome that carries the H2A variant (H2Av) at the promoter region. Upon developmental triggers or heat shock-induced phosphorylation of H2Av, PARP-1 is activated (***Kotova et al., 2011***). Activated PARP-1 fosters a transcription start site (TSS) that is more accessible, thereby enabling the binding of RNA-Polymerase II (Pol II) and initiation of transcription (***Nikolaou et al., 2017***). Following this, the distribution of PARP-1 is further enhanced by active mono-methylated forms of H4K20, H3K4, H3K36, or H3K9. Each of these histone modifications can interchangeably facilitate this function. The spreading of PARP-1 to the gene body contributes to the loosening of chromatin in this region (***Tulin and Spradling, 2003***; ***Estève et al., 2022***; ***Petesch and Lis, 2012***; ***Rickels et al., 2017***). Consequently, this facilitates the transition of Pol II into a productive elongation phase, leading to the generation of a mature transcript.

The online version of this article includes the following source data for figure 5:

**Source data 1.** Original file for the Western blot analysis in *Figure 5B*.

**Source data 2.** This file contains *Figure 5B* and original scans of the relevant Western blot analysis with highlighted bands and sample labels.

(DSB) DNA damage repair, all biological processes in which PARP-1 plays a role (***Kraus and Hottiger, 2013***; ***Oda et al., 2010***; ***Beck et al., 2012***; ***Ji and Tulin, 2010***; ***Khoury-Haddad et al., 2014***; ***Tuzon et al., 2014***; ***Yang et al., 2013***). Hence, histone modifications such as H4K20me1 may act to regulate the activity of key chromatin remodelers such as PARP-1.

A recent study demonstrated that in human cells overexpressing PARP-1, PR-SET7/SET8 is degraded, whereas depletion of PARP-1 leads to an increase in PR-SET7/SET8 levels (***Estève et al., 2022***). However, in our study involving *parp-1* mutant in *Drosophila* third-instar larvae revealed a nuanced scenario: we detected a minor but not significant reduction in both PR-SET7 RNA and protein levels (***Figure 3—figure supplements 2 and 3***). This outcome stands in stark contrast to the previous study's findings. The discrepancy could be due to the distinct experimental approaches used: the previous research focused on mammalian cells and in vitro experiments, whereas our study examined

the functions of PARP-1 in whole *Drosophila* third-instar larvae during development. Consequently, while PARP-1 may cooperate with PR-SET7 in the context of *Drosophila* development, it could exhibit antagonistic roles against PR-SET7 in specific cell lines and under certain biological or developmental conditions.

Our transcriptomic analysis from *pr-set7[20]* and *parp-1[C03256]* mutant third-instar larvae implies that PR-SET7/H4K20me1 might be crucial for PARP-1's role in transcriptional regulation. Remarkably, differentially expressed genes (DEGs) in *pr-set7[20]* and *parp-1[C03256]* mutants exhibited a high degree of correlation (*Figure 3A*), although we noticed a minimal alteration in gene expression where PARP-1 was co-localized with H4K20me1 across the genome (*Figure 3C and D*). This suggests that H4K20me1 might only be essential for PARP-1-mediated binding and gene regulation at a select group of genes, specifically during metabolic gene repression and developmental gene activation. Our results also propose that PARP-1 and PR-SET7/H4K20me1 might have separate roles in gene regulation at co-enriched genes.

Notably, genes co-enriched with PARP-1 and H4K20me1, and are upregulated in both *parp-1[C03256]* and *pr-set7[20]* mutants, are predominantly metabolic genes exhibiting high expression levels under wild-type conditions and a high occupancy of RNA-polymerase II both at their promoter region and gene body (*Figure 3—figure supplement 4*). In our previous studies, we discovered that PARP-1 plays a crucial role in repressing highly active metabolic genes during the development of *Drosophila* by binding directly to their loci (*Bamgbose and Tulin, 2024*; *Bamgbose et al., 2023*). Also, PARP-1 is required for maintaining optimum glucose and ATP levels at the third-instar larval stage (*Bamgbose and Tulin, 2024*). During *Drosophila* development, repression of metabolic genes is crucial for larval to pupal transition (*Arbeitman et al., 2002*; *White et al., 1999*). This repression is linked to the reduced energy requirements as the organism prepares for its sedentary pupal stage (*Arbeitman et al., 2002*; *Nishimura, 2020*). Remarkably, our observations indicate a notable affinity of PARP-1 for binding to the gene bodies of these metabolic genes (*Bamgbose and Tulin, 2024*), suggesting a direct involvement of PARP1 in their regulation. Nonetheless, it remains plausible that certain genes may be indirectly regulated by PARP1 through intermediary transcription factors.

Our data indicates that in both *parp-1* and *pr-set7* mutant animals, there was a preferential repression of metabolic genes at sites where PARP-1 and H4K20me1 are co-bound (*Figure 3E*), while these metabolic genes are highly active during third-instar larval stage (*Figure 3—figure supplement 4*). Thus, we propose that the presence of H4K20me1 may be essential for the binding of PARP-1 at these gene bodies, contributing to their repression. Importantly, this mechanism of gene repression has broader developmental implications. As earlier stated, mutant animals lacking functional PARP-1 and PR-SET7 undergo developmental arrest during larval to pupal transition. This arrest could be directly linked to the disruption of the normal metabolic gene repression during development. Without the repressive action of PARP-1 and PR-SET7, key metabolic processes might remain unchecked, leading to metabolic imbalances that are incompatible with the normal progression to the pupal stage.

In this study, we have demonstrated that PARP-1 is essential for the activation of *hsp22, hsp23,* and *hsp83* during heat shock (*Figure 4A*). Importantly, the expression of these heat shock genes significantly decreased in *pr-set7[20]* mutants following heat shock (*Figure 4A*), which shows a role for PR-SET7/H4K20me1 in the heat shock response. We have also observed that H4K20me1 regulation is dynamic in the gene bodies of PARP-1- and PR-SET7-dependent heat shock genes. Specifically, at genes that show low or no H4K20me1 enrichment prior to heat shock, H4K20me1 levels increase post-heat shock (*Figure 4B and D*). Conversely, at genes with high levels of H4K20me1, the level of H4K20me1 decreases following heat shock (*Figure 4C and D*). Previous studies have shown that H4K20me1 occupancy can either inhibit or facilitate transcriptional elongation, depending on the biological context (*Nikolaou et al., 2017*; *Vakoc et al., 2006*; *Tanaka et al., 2017*; *Veloso et al., 2014*; *Wang et al., 2015*). As such, we propose that for group A genes, which show a significant increase in H4K20me1 enrichment after heat shock, H4K20me1 may facilitate the spread of PARP-1 in their gene bodies, thus enhancing transcriptional activation. In contrast, the removal of H4K20me1 at group B genes, which exhibit a significant reduction of H4K20me1 at their gene bodies, may enable transcriptional elongation. Thus, an unidentified H4K20me1 demethylase and PR-SET7's non-catalytic functions may be required to facilitate transcriptional activation of group B heat shock genes during heat stress.

Another plausible explanation could be that the recruitment of PARP-1 to group B genes loci promotes H4 dissociation and then leads to a reduction of H4K20me1. However, our findings suggest an alternative interpretation: the decrease in H4K20me1 at group B genes during heat shock does not seem to impede PARP-1's role in transcriptional activation, (*Figure 4A, C and E*). Rather than disrupting PARP-1 function, we propose that this reduction in H4K20me1 may signify a regulatory shift in chromatin structure, priming these genes for transcriptional activation during heat shock, with PARP-1 playing an independent facilitating role. Moreover, existing studies have highlighted the dual role of H4K20me1, acting as a promoter of transcription elongation in certain contexts and as a repressor in others (*Barski et al., 2007*; *Nikolaou et al., 2017*; *Tanaka et al., 2017*; *Veloso et al., 2014*; *Fuchs et al., 2014*; *Abbas et al., 2010*; *Kapoor-Vazirani and Vertino, 2014*; *Shoaib et al., 2021*; *Gungi et al., 2023*). The elevated enrichment of H4K20me1 in group B genes under normal conditions may indicate a repressive state that requires alleviation for transcriptional activation. Additionally, we cannot discount the possibility of unique regulatory functions associated with PR-SET7, extending beyond its recognized role as a histone methylase. Non-catalytic activities and potential interactions with non-histone substrates might contribute to the nuanced control exerted by PR-SET7 on group B genes during heat stress (*Shi et al., 2007*; *Takawa et al., 2012*). Furthermore, our exploration of *pr-set7[20]* and *parp-1[C03256]* mutants reveals distinct roles for PARP-1 and H4K20me1 in modulating gene expression (*Figure 3E*). This reinforces the notion that the interplay between PR-SET7 and PARP-1 involves a multifaceted regulatory mechanism. Understanding the intricate relationship between these molecular players is crucial for elucidating the complexities of gene expression modulation under heat stress conditions.

Notably, *Drosophila* embryos expressing an unmodifiable H4K20 (H4K20A) and a catalytically-inactive *trr* mutant (monomethylase of H3K4) were still able to mature into adult flies (*Karachentsev et al., 2005*; *Rickels et al., 2017*), demonstrating that H3K4me1 and H4K20me1 are dispensable for transcription during development. This finding contrasts our hypothesis that H4K20me1 or H3K4me1 might modulate PARP-1 binding at PARP-1-dependent genes during development. Importantly, our data also revealed that PARP-1 binds to H3K36me1 and H3K9me1. Consequently, we suggest that H3K36me1 and H3K9me1 could potentially substitute the role of H3K4me1 and H4K20me1 in controlling PARP-1 binding at PARP-1 dependent genes (*Figure 5C*).

Finally, highly transcribed genes have been reported to present a high turnover of mono-methylated modifications, maintaining a state of low methylation (*Chory et al., 2019*). Moreover, our previous study revealed that PARP1 preferentially binds to highly active genes (*Bamgbose and Tulin, 2024*). Consequently, our findings suggest an active involvement of PARP-1 in the turnover process to maintain an active chromatin environment. This proposed mechanism unfolds in the following steps: (1) PARP-1 selectively binds to mono-methylated active histone marks associated with highly transcribed genes. (2) Upon activation, PARP-1 undergoes automodification and subsequently disengages from chromatin, facilitating the reassembly of nucleosomes carrying the mono-methylated marks. (3) The enzymatic action of Poly(ADP)-ribose glycohydrolase (PARG) cleaves pADPr, restoring PARP-1's binding affinity to mono-methylated active histone marks. This proposed hypothesis is consistent with existing research conducted across various model organisms, including mice, *Drosophila*, and Humans (*Tulin and Spradling, 2003*; *Krishnakumar and Kraus, 2010*; *Ji and Tulin, 2010*; *Liu and Kraus, 2017*; *Tulin et al., 2002*; *Muthurajan et al., 2014*). Notably, previous studies have consistently demonstrated that PARP-1 predominantly associates with highly expressed genes and plays a crucial role in mediating nucleosome dynamics and assembly. Thus, our proposed model provides a molecular framework that may contribute to understanding the relationship between PARP-1 and the epigenetic regulation of gene expression.

# Materials and methods
## *Drosophila* strains and genetics

Flies were cultured on standard cornmeal-molasses-agar media at 25 °C. The transgenic stock pP{w[1], UAST::PARP1-EYFP} has been previously described (*Tulin et al., 2002*). The *parp-1[C03256]* hypomorph mutant was generated in a single pBac-element mutagenesis screen (*Artavanis-Tsakonas, 2004*). *Parp-1[C03256]* significantly lowers the level of PARP-1 RNA and protein level (*Kotova et al., 2010*) but also significantly diminishes the level of pADPr (*Kotova et al., 2011*). The *pr-set7[20]* null mutant was

generously provided by Dr. Ruth Steward (*Karachentsev et al., 2005*). The *pr-set7²⁰* null mutant was validated in *Karachentsev et al., 2005* and we confirmed that this mutant abolishes PR-SET7 RNA and protein level but also leads to the absence of H4K20me1 (*Figure 3—figure supplement 1*). Wandering third-instar larvae was used for all experiments. *w¹¹¹⁸* strains were used as controls for ChIP-seq and RNA-seq and termed WT as appropriate. We used TM6B and TM3-GFP to isolate *parp-1^C03256* and *pr-set7²⁰* homozygous mutants, respectively.

## Histone peptide array

The Modified Histone peptide array (Active Motif) was blocked by overnight incubation in the tween 20 containing tris buffered saline (TTBS) buffer (10 mm Tris/HCl pH 8.3, 0.05% Tween-20 and 150 mm NaCl) containing 5% non-fat dried milk at 4 °C. The membrane was then washed once with the TTBS buffer and incubated with 4.0 gm PARP-1 (Trevigen) in PARP binding buffer (10 mm Tris-HCl, pH 8, 140 mm NaCl, 3 mm DTT, and 0.1% Triton X-100) at room temperature for 1 h. The membrane was washed in the TTBS buffer and incubated with anti-PARP antibody (SERUTEC at 1:500 dilution) for 1 hr at room temperature in blocking buffer (5% milk in TTBS). The unbound antibody was washed three times with TTBS, and the membrane was incubated with horseradish peroxidase-conjugated anti-mouse antibody (Sigma, 1:2500) in TTBS for 1 hr at room temperature. Finally, the membrane was submerged in ECL developing solution (GE Healthcare) and the image was captured on X-ray film. Typical exposure times were 0.5–2 min. The images were analyzed using an in-house program (Array Analyze, available at here).

## *Drosophila* salivary gland polytene chromosome immunostaining

Preparation and immunostaining of polytene chromosome squashes were performed exactly as described (*Johansen et al., 2009*). The primary antibody used was anti-GFP (Living Colors, #JL-8, 1:100), H3K4me1 (Abcam, 1:50) and H4K20me1 (Abcam, 1:100) and the secondary antibody used was goat anti-mouse Alexa-488 (Molecular Probes, 1:1500) and goat-anti-rabbit Alexa-568 (Molecular Probes, 1:1500). DNA was stained with TOTO3 (1:3000). Slides were mounted in Vectashield (Vector Laboratories, Burlingame, CA).

## Chromatin immunoprecipitation and sequencing

PARP-1-YFP ChIP-seq has been previously described (*Bamgbose et al., 2023*). Briefly, 75 third-instar larvae were collected, washed with 1 ml 1 X PBS, and homogenized in lysis buffer (200 µl 1 X protease inhibitor cocktail, 250 µl PMSF, 800 µl 1 X PBS, 1 µl Tween 20). Crosslinking was achieved by adding 244.5 µl of 11% formaldehyde for a final concentration of 1.8% and quenched with 500 mM glycine. The pellet was resuspended in sonication buffer (0.5% SDS, 20 mM Tris pH 8.0, 2 mM EDTA, 0.5 mM EGTA, 0.5 mM PMSF, 1 X protease inhibitor cocktail) and sonicated to fragment chromatin. The supernatant was then collected after pelleting. This was pre-cleared, incubated with anti-GFP antibody (TP-401, Torrey Pines Biolabs), and immunoprecipitated chromatin was collected using Protein A agarose beads. Sequential washing was performed using low salt buffer, high salt buffer, LiCl wash, and TE buffer washers. Bound chromatin was eluted using 250 µl ChIP elution buffer (1% SDS, 100 mM NaHCO3) and reverse-crosslinked. The eluates were treated with RNase A and proteinase K, and DNA was extracted via phenol-chloroform extraction and ethanol precipitation. Sequencing was carried out at Novogene.

## ChIP-seq analysis

The quality of FASTQ files (raw reads) were checked using FastQC (version. 0.11.9) and adapters were removed with fastp (*Chen et al., 2018*). Trimmed FASTQ files were aligned to the *Drosophila* genome (dm6) using Bowtie2 to generate bam files (*Langmead and Salzberg, 2012*). Unmapped and low-quality reads were discarded from bam files ( ≤20 mapQuality) using BamTools (*Barnett et al., 2011*). Duplicate reads were identified and removed from mapped reads using Picard Mark-Duplicates (Picard 2.8.0; RRID:SCR_006525 ;http://broadinstitute.github.io/picard/). Deeptools MultiBamSummary was used to determine reproducibility of ChIP-seq reads. MACS2 was used to call peaks against pooled Input/control using default settings except narrowPeaks were called for PARP-1 and broadPeaks were called for H3K4me1 (gapped), H3K9me1, H4K20me1, H3K36me1, H3K9me2, H3K9me3. Peaks were annotated to genomic features with ChIPseeker (*Yu et al., 2015*).

Pairwise correlation of peaks was determined using Intervene (*Khan and Mathelier, 2017*). MACS2 bedGraph pileups were used to generate normalized coverage of ChIP-seq signals using Deeptools bigWigCompare by computing the ratio of the signals (IP vs Control/Input) using a 50 bp bin size. Deeptools multiBigwigSummary was used to determine genome-wide signal correlation using a 10 kb bin size. Deeptools plotHeatmap was used to create gene-centric enrichment profiles using scaled region mode. Gene clusters were determined via K means function (n=3) using Deeptools suite (*Ramírez et al., 2016*).

To compare H4K20me1 and PARP-1 occupancy in Human K562 cells (*Figure 3—figure supplement 5*), CrossMap was used to convert H4K20me1 bigwig from Hg19 to Hg38 to match PARP-1 bigwig. Deeptools were used to generate and plot enrichment of PARP-1 and H4K20me1 at PARP-1 gene clusters.

## Mononucleosome chromatin immunoprecipitation

We collected 60 WT wandering *Drosophila* third-instar larvae and rinsed them in 1 X PBS twice. For heat shock experiments, larvae were heat-shocked in a 1.5 ml DNA LoBind tube for 15 min and 30 min in a water bath at 37 °C. Larvae were then homogenized using a pellet pestle homogenizer in 500 µl ice-cold buffer A1 60 mM KCl, 15 mM NaCl, 4 mM MgCl2, 15 mM 4-(2-Hydroxyethyl)piperazine-1-ethanesulfonic acid (HEPES) (pH 7.6), 0.5% Triton X-100, 0.5 mM DTT, 1X EDTA (Ethylene Diamine Tetraacetic Acid)-free protease inhibitor cocktail (Roche 04693132001). The larvae were then cross-linked by adding formaldehyde to a 1.8% concentration and incubating for 5 min at RT on a rotator. Crosslinking was stopped by adding glycine (final concentration was 0.125 M) at room temperature for 5 min on a rotator. The mixture was centrifuged at 2000 g for 5 min at 4 °C, and the supernatant was discarded. The pellet was washed as follows: once with 500 µl A1 buffer, and once with 500 µl A2 buffer (140 mM NaCl, 15 mM HEPES (pH 7.6), 1 mM EDTA, 0.5 mM ethylene glycol tetraacetic acid (EGTA), 1% Triton X-100, 0.1% sodium deoxycholate, 0.5 mM DTT, 1X EDTA free protease inhibitor cocktail (Roche 04693132001)). For each wash, the tube was shaken for 1 min and centrifuged as before, and the supernatant was discarded. The pellet was resuspended in 500 µl A2 buffer + 0.1% sodium dodecyl sulfate (SDS) and incubated on a rotating wheel at 4 °C for 10 min. Then the mixture was centrifuged at 16,000 g for 5 min at 4 °C and the supernatant was discarded. The pellet was washed with 500 µl MNase digestion buffer (10 mM Tris–HCl (pH 7.5), 15 mM NaCl, 60 mM KCl, 1 mM CaCl2, 0.15 mM spermine, 0.5 mM spermidine, 1X EDTA free protease inhibitor cocktail) and centrifuged at 16,000 g for 10 min at 4 °C. The nuclei were resuspended in 500 µl MNase digestion buffer and 400 U MNase (Worthington, LS004797) was added and incubated at 37 °C for 20 min. The MNase digestion was terminated on ice by adding EDTA to a final concentration of 10 mM and kept on ice for 10 min. The mixture was centrifuged at 16,000 g for 10 min at 4 °C. The supernatant was discarded. The pellet was resuspended in 500 µl A3 buffer (140 mM NaCl, 15 mM HEPES (pH 7.6), 1 mM EDTA, 0.5 mM EGTA, 1% Triton X-100, 0.1% sodium deoxycholate, 0.1% SDS, 1X EDTA free protease inhibitor cocktail) and incubated for 30 min on a rotator at 4 °C. The mixture was then centrifuged at 16,000 g for 15 min at 4 °C. The supernatant containing mononucleosomal fragments was transferred to a new DNA LoBind tube, then 20 µl of mixture was used to check fragment size prior to immunoprecipitation. The mixture was then made up to 2 ml with IP buffer (0.5% SDS, 20 mM Tris, pH 8.0, 2 mM EDTA, 0.5 mM EGTA, 0.5 mM PMSF, protease inhibitor cocktail). The mononucleosomal chromatin was pre-cleared and 800 µl was incubated with 5 µl anti-H4K20me1 antibody (Abcam, 9051) or 5 µl Rabbit IgG control at 4 °C overnight, and 80 µl was kept as input. The immunoprecipitated chromatin was then collected with pre-washed Protein A agarose beads for 2 hr. The beads were sequentially washed with the following buffers: 1 low salt buffer wash (0.1% SDS, 1% Triton X-100, 2 mM EDTA, 20 mM Tris-HCL pH 8.0, 150 mM NaCl), three high salt buffer washes (0.1% SDS, 1% Triton X-100, 2 mM EDTA, 20 mM Tris-HCL pH 8.0, 500 mM NaCl), one LiCL wash (2 mM EDTA, 20 mM Tris-HCl pH 8.0, 0.25 M LiCl, 1% NP-40), and two TE buffer washers before elution. Bound chromatin on beads was eluted twice at room temperature using 250 µl of freshly prepared ChIP elution buffer (1% SDS, 100 mM NaHCO3) for 15 min and reverse-crosslinked overnight. The eluates were then treated with RNase A and proteinase K prior to DNA extraction via phenol-chloroform extraction and ethanol precipitation.

## RNA sequencing

RNA was isolated from 10 wandering third-instar larvae (three biological replicates per genotype; WT, *pr-set7[20]*, and *parp-1[C03256]*) using RNeasy lipid tissue mini kit (Qiagen). RNA samples were flash-frozen in liquid nitrogen and sent to Novogene for library preparation and sequencing. mRNA was purified from total RNA via poly-T oligo beads. Libraries were prepared using an Ultra II RNA library kit (NEB) and samples were sequenced on the NovaSeq 6000 platform (Illumina) at Novogene.

## RNA-seq analysis

Paired-end reads were quality-checked using FastQC and reads were mapped to the *Drosophila* genome (dm6) using RNA STAR (*Dobin et al., 2013*). Reads per annotated gene was counted using featureCounts (*Liao et al., 2014*). Differential expression analysis was performed with DESeq2 (*Love et al., 2014*), with Log2 fold change of at least 1 (absolute) considered significant (FDR <0.05).

## Quantitative RT-PCR

For heat shock experiments, 10 third-instar larvae were heat-shocked in 1.5 ml tube in a water bath at 37 °C for 30 min. Total RNA was then isolated for three biological replicates using an RNeasy Lipid Tissue Mini kit (Qiagen). DNA contamination was removed using TURBO DNA-free (Ambion). RNA concentration was measured using an Agilent 2100 BioAnalyzer (Agilent Technologies) in combination with an RNA 6000 Nano LabChip (Agilent Technologies). Real-time PCR assays were run using StepOnePlus Real-Time PCR System (Applied Biosystems).

Primers used for *hsp70* ChIP-qPCR were obtained from *Petesch and Lis, 2008*; *Berson et al., 2017*. Primers used for *hsp22, hsp23, hsp68,* and *hsp83* qPCR were obtained from *Love et al., 2014*. Primer sequences are listed in *Supplementary file 2*.

## *pr-set7[20]* mutants validation using qRT-PCR

Total RNA was isolated from 10 third-instar larvae for three biological replicates using an RNeasy Lipid Tissue Mini kit (Qiagen). DNA contamination was removed using TURBO DNA-free (Ambion). RNA concentration was measured using an Agilent 2100 BioAnalyzer (Agilent Technologies) in combination with an RNA 6000 Nano LabChip (Agilent Technologies). Real-time PCR assays were run using StepOnePlus Real-Time PCR System (Applied Biosystems). The following primers were used: *pr-set7* (Forward) CGCACAATAGGAGTTCCC (Reverse) CCTCATCGTCCAGTTTCAG and *rp49* (Forward) GCTAAGCTGTCGCACAAAT (Reverse) GAACTTCTTGAATCCGGTGGG.

## PARP-1-EYFP re-localization in chromatin during heat shock

Larvae were grown at 20 °C. Heat shock was performed by placing the third instar wandering larvae individually in a 1.5 ml Eppendorf tube with a small hole in the cap to let the larvae breath. The tube was placed in a Marry bath at 37 °C for 30 min. After heat shock, the larvae were briefly washed in 1 X PBS and then immediately dissected in 1 X supplemented Grace's insect media (Gibco). Salivary glands were mounted in 1 X supplemented Grace's insect media (Gibco) containing Draq-5 DNA dye (Life Technologies) at a 1/5000 dilution.

## Western blot analysis

The following antibodies were used for immunoblotting assays: anti-PR-SET7 (Rabbit, 1:1000, Novus Biologicals, 44710002), anti-H4K20me1 (Rabbit, 1:1000, Abcam, ab9051), anti-H4 (Mouse, 1:1000, Abcam, ab31830), anti-pADPr (Mouse monoclonal, 1:500, 10 H - sc-56198, Santa Cruz), anti-H3 (Rabbit polyclonal, 1/1000, FL-136 sc-10809 Santa Cruz).

## Gene ontology (GO)

Gene ontology terms were determined using g:profiler (FDR <0.05) (*Raudvere et al., 2019*).

## Statistical analysis

Results were analyzed using the indicated statistical test in GraphPad Prism (9.4.0). Statistical significance of ChIP-seq peaks were determined using MACS2. Q-value cutoffs for RNA-seq and GO analysis were determined with DESeq2 and G:profiler, respectively.

## Acknowledgements

We thank Dr. Ruth Steward for gifting us *pr-set7*[20] null mutants. We thank Yaroslava Karpova who made the schemes of *Figures 1A and 5C*. This study was supported by the Department of Defense grants PC160049 and the National Science Foundation MCB-2231403 to AVT; International Training Scholarship by the American Society for Cell Biology through the International Federation for Cell Biology to GB.

## Additional information

### Funding

| Funder | Grant reference number | Author |
| --- | --- | --- |
| National Science Foundation | MCB-2231403 | Alexei Tulin |
| Department of Defense | PC160049 | Alexei Tulin |

The funders had no role in study design, data collection and interpretation, or the decision to submit the work for publication.

### Author contributions

Gbolahan Bamgbose, Conceptualization, Data curation, Formal analysis, Validation, Investigation, Methodology, Writing - original draft; Guillaume Bordet, Formal analysis, Validation, Investigation, Visualization, Methodology, Writing - review and editing; Niraj Lodhi, Conceptualization, Data curation, Validation; Alexei Tulin, Conceptualization, Resources, Data curation, Formal analysis, Supervision, Funding acquisition, Validation, Investigation, Visualization, Methodology, Project administration, Writing - review and editing

### Author ORCIDs

Gbolahan Bamgbose ⓘD http://orcid.org/0009-0000-5406-1894
Guillaume Bordet ⓘD http://orcid.org/0000-0002-1346-0984
Alexei Tulin ⓘD http://orcid.org/0000-0001-9618-4243

Reviewer #2 (Public Review): https://doi.org/10.7554/eLife.91482.4.sa1
Author response https://doi.org/10.7554/eLife.91482.4.sa2

## Additional files

### Supplementary files

• Supplementary file 1. Data of the PARP-1 histone peptide array. Sheet 1 displays raw and processed data for each modification tested. Column 1 displays the peptide number. Column B displays the location on the plate. Column C displays the name of the Histone tail. Columns D to G displays the modification(s) present on the histone tail. Columns H and I display the intensity recorded for the peptide in Array 1 and 2, respectively. Column J displays the average intensity, calculated based on values from columns H and I. Column K displays the standard error of mean based on values from columns H and I. Sheet 2 displays the specificity factor for each histone modification. Column A displays the rank of the modification (based on their specificity factor). Column B displays the name of the histone modification. Columns C and D display the number of dots of the array that contain the modification (column C) or that does not contain the modification (column D). Columns E and F display the average intensity of the positive (E) or negative (F) dots. Column G displays the specificity factor of the histone modification.

• Supplementary file 2. Primers used in this study. This supplementary file displays the list of primers used for qPCR (sheet 1) or for ChIP-qPCR (sheet 2) in this study.

• MDAR checklist

## Data availability

All data generated or analyzed during this study are included in this article. No custom code was generated in this study. ChIP-seq and RNA-seq data generated in this study are available on the GEO database (Accession no: GSE217730 and GSE222877). The following public *Drosophila* third-instar larvae ChIP-seq (ENCODE) were used in this study: GSE47282 (H3K4me1), GSE47254 (H4K20me1), GSE47249 (H3K36me1), GSE47289 (H3K9me1), GSE47260 (H3K9me2) and GSE47258 (H3K9me3), GSE15292 (RNA-Polymerase II). Finally, the following public Human K562 cells ChIP-seq (ENCODE) were used in this study: GSE206022 (PARP-1 CUT&Tag), GSE29611 (H4K20me1).

The following datasets were generated:

| Author(s) | Year | Dataset title | Dataset URL | Database and Identifier |
|---|---|---|---|---|
| Tulin A | 2023 | Role of PARP-1 in transcriptional regulation during development | https://www.ncbi.nlm.nih.gov/geo/query/acc.cgi?acc=GSE217730 | NCBI Gene Expression Omnibus, GSE217730 |
| Tulin A | 2023 | Differential binding of PARP-1 domains | https://www.ncbi.nlm.nih.gov/geo/query/acc.cgi?acc=GSE222877 | NCBI Gene Expression Omnibus, GSE222877 |

The following previously published datasets were used:

| Author(s) | Year | Dataset title | Dataset URL | Database and Identifier |
|---|---|---|---|---|
| ENCODE Project Consortium | 2011 | Histone Modifications by ChIP-seq from ENCODE/Broad Institute | https://www.ncbi.nlm.nih.gov/geo/query/acc.cgi?acc=GSE29611 | NCBI Gene Expression Omnibus, GSE29611 |
| Barshad G, Lewis JJ, Chivu AG, Abuhashem A | 2022 | RNA polymerase II and PARP1 shape enhancer-promoter contacts [CUT&Tag] | https://www.ncbi.nlm.nih.gov/geo/query/acc.cgi?acc=GSE206022 | NCBI Gene Expression Omnibus, GSE206022 |
| Nègre N, Brown CD, Ma L, Bristow CA | 2009 | *Drosophila* at different time points of development: ChIP-chip, ChIP-seq, RNA-seq | https://www.ncbi.nlm.nih.gov/geo/query/acc.cgi?acc=GSE15292 | NCBI Gene Expression Omnibus, GSE15292 |
| Karpen G, Elgin S, Gortchakov A, Shanower G, Tolstorukov M, Kharchenko P, Kuroda M, Pirrotta V, Park P, Minoda A, Riddle N, Schwartz Y, Alekseyenko A, Kennedy C | 2013 | H3K9me3 (new lot).D.mel 3rd Instar Larvae Nuclei. Solexa | https://www.ncbi.nlm.nih.gov/geo/query/acc.cgi?acc=GSE47258 | NCBI Gene Expression Omnibus, GSE47258 |
| Karpen G, Elgin S, Gortchakov A, Shanower G, Tolstorukov M, Kharchenko P, Kuroda M, Pirrotta V, Park P, Minoda A, Riddle N, Schwartz Y, Alekseyenko A, Kennedy C | 2013 | H3K9me2 antibody2.D.mel 3rd Instar Larvae Nuclei. Solexa | https://www.ncbi.nlm.nih.gov/geo/query/acc.cgi?acc=GSE47260 | NCBI Gene Expression Omnibus, GSE47260 |

*Continued on next page*

*Continued*

| Author(s) | Year | Dataset title | Dataset URL | Database and Identifier |
|---|---|---|---|---|
| Karpen G, Elgin S, Gortchakov A, Shanower G, Tolstorukov M, Kharchenko P, Kuroda M, Pirrotta V, Minoda A, Riddle N, Schwartz Y, Alekseyenko A, Kennedy C | 2013 | H3K9me1 (new lot 2).D.mel 3rd Instar Larvae Nuclei. Solexa | https://www.ncbi. nlm.nih.gov/geo/ query/acc.cgi?acc= GSE47289 | NCBI Gene Expression Omnibus, GSE47289 |
| Karpen G, Elgin S, Gortchakov A, Shanower G, Tolstorukov M, Kharchenko P, Kuroda M, Pirrotta V, Park P, Minoda A, Riddle N, Schwartz Y, Alekseyenko A, Kennedy C | 2013 | H3K36me1.D.mel 3rd Instar Larvae Nuclei.Solexa | https://www.ncbi. nlm.nih.gov/geo/ query/acc.cgi?acc= GSE47249 | NCBI Gene Expression Omnibus, GSE47249 |
| Karpen G, Elgin S, Gortchakov A, Shanower G, Tolstorukov M, Kharchenko P, Kuroda M, Pirrotta V, Park P, Minoda A, Riddle N, Schwartz Y, Alekseyenko A, Kennedy C | 2013 | H4K20me1 (3rd lot).D.mel 3rd Instar Larvae Nuclei. Solexa | https://www.ncbi. nlm.nih.gov/geo/ query/acc.cgi?acc= GSE47254 | NCBI Gene Expression Omnibus, GSE47254 |
| Kharchenko P, Kuroda M, Pirrotta V, Park P, Minoda A, Riddle N, Schwartz Y, Alekseyenko A, Kennedy C | 2013 | H3K4me1 (ab8895 new lot 2).D.mel 3rd Instar Larvae Nuclei.Solexa | https://www.ncbi. nlm.nih.gov/geo/ query/acc.cgi?acc= GSE47282 | NCBI Gene Expression Omnibus, GSE47282 |

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
