## [Editor Report · eLife assessment]

This **valuable** study presents **convincing** evidence for an association between PARP-1 and H4K20me1 in transcriptional regulation, supported by biochemical and ChIP-seq analyses. The work contributes significantly to our understanding of how Parp1 associates with target genes to regulate their expression.

---

## [Referee Report · Reviewer #2 (Public Review)]

Summary:

This study from Bamgbose et al. identifies a new and important interaction between H4K20me and Parp1 that regulates inducible genes during development and heat stress. The authors present convincing experiments that form a mostly complete manuscript that significantly contributes to our understanding of how Parp1 associates with target genes to regulate their expression.

Strengths:

The authors present 3 compelling experiments to support the interaction between Parp1 and H4K20me, including:

(1) PR-Set7 mutants remove all K4K20me and phenocopy Parp mutant developmental arrest and defective heat shock protein induction.

(2) PR-Set7 mutants have dramatically reduced Parp1 association with chromatin and reduced poly-ADP ribosylation.

(3) Parp1 directly binds H4K20me in vitro.

---

## [Author Response]

The following is the authors’ response to the previous reviews

We extend our sincere gratitude for the invaluable comments provided by the reviewers and yourself, along with the constructive suggestions to enhance the quality of our manuscript. In response to this invaluable feedback, we have diligently revised and resubmitted our paper as an article, introducing five primary figures, seven supplementary figures, and two supplementary data files. Importantly, this work represents a significant contribution to the field, presenting novel findings for the first time without any prior publication.

Within the enclosed document, we have provided a comprehensive response to the editor and reviewer comments, addressing each point meticulously and specifically. We extend our heartfelt thanks to the reviewers and yourself for your diligent examination of our manuscript and for offering insightful recommendations.

In our latest revision, we have taken great care to address every comment, ensuring that we clarify the manuscript and provide robust evidence where required. We have meticulously highlighted the modifications within the manuscript in yellow for your convenience, while also including the modifications made in response to each specific comment. The primary focus of these revisions was to provide additional context regarding the relationship between PARP-1 and mono-methylated histones. Substantial modifications were made to our discussion section to address this point.

Another concern raised was regarding the discrepancy in the relationship of PR-SET7 and PARP-1 between our study and the recent study by Estève et al. (PMID: 36434141). We have revised the results and discussion sections to discuss this concern.

Addressing Reviewer 2’s concern about the potential indirect role of PARP1 in the regulation of some metabolic genes despite its direct binding to loci coding for metabolic genes we revised the discussion section to highlight this possibility.

Enclosed, you will find a detailed, point-by-point response to each of the editor’s and reviewers' comments, showcasing our commitment to addressing their concerns with precision.

We firmly believe that our revisions successfully resolve all the concerns raised by the editor and the reviewers, and we are confident that this improved version of our manuscript contributes significantly to the scientific discourse. Once again, we thank you for considering our work, and please feel free to contact me if you require any additional information.

In the revised manuscript, most of the concerns raised by the reviewers have been addressed satisfactorily. However, as suggested by reviewer#2, it would have been more significant, if the PARP1-mediated reading of global mono-methylation of histone could be addressed. At least the mechanisms of selectivity of PARP1 need further convincing discussion.

We thank the editor for their valuable comments. We have extended our discussion section to discuss in more detail the relationship between PARP1 and mono-methylated histones. In our refined Discussion section, we have endeavored to articulate more clearly how PARP-1 may be selectively recruited to active chromatin domains through its interaction with mono-methylated histone marks. We propose a model wherein PARP-1 actively participates in the turnover process, contributing to the maintenance of an active chromatin environment. This mechanism entails PARP-1 selectively binding to mono-methylated active histone marks associated with highly transcribed genes. Upon activation, PARP-1 undergoes automodification, leading to its release from chromatin and facilitating the reassembly of nucleosomes carrying the mono-methylated marks. Subsequently, the enzymatic action of Poly(ADP)-ribose glycohydrolase (PARG) cleaves pADPr, enabling the restoration of PARP-1's binding affinity to mono-methylated active histone marks. This proposed hypothesis is consistent with existing research across various model organisms and aligns with the known association of PARP-1 with highly expressed genes, as well as its role in mediating nucleosome dynamics and assembly.

Our modified Discussion section unfolds as follows:

"Finally, highly transcribed genes have been reported to present a high turnover of mono-methylated modifications, maintaining a state of low methylation (50). Moreover, our previous study revealed that PARP1 preferentially binds to highly active genes (34). Consequently, our findings suggest an active involvement of PARP-1 in the turnover process to maintain an active chromatin environment. This proposed mechanism unfolds in the following steps: (1) PARP-1 selectively binds to mono-methylated active histone marks associated with highly transcribed genes. (2) Upon activation, PARP-1 undergoes automodification and subsequently disengages from chromatin, facilitating the reassembly of nucleosomes carrying the mono-methylated marks. (3) The enzymatic action of Poly(ADP)-ribose glycohydrolase (PARG) cleaves pADPr, restoring PARP-1's binding affinity to mono-methylated active histone marks. This proposed hypothesis is consistent with existing research conducted across various model organisms, including mice, *Drosophila*, and Humans (7, 24, 30, 51-53). Notably, previous studies have consistently demonstrated that PARP-1 predominantly associates with highly expressed genes and plays a crucial role in mediating nucleosome dynamics and assembly. Thus, our proposed model provides a molecular framework that may contribute to understanding the relationship between PARP-1 and the epigenetic regulation of gene expression."

We trust that these revisions effectively address the editor’s comment and enhance the overall strength and clarity of our manuscript.

Furthermore, recent developments in the area are omitted, as an important publication hasn't been discussed anywhere in the work (PMID: 36434141).

We appreciate the editor's thorough review of our revised manuscript and the responses to the previous reviewer's comments. To address this important concern, we have carefully investigated the levels of PR-SET7 in *parp1* hypomorphic conditions.

Supplemental Fig. S4 and S5 demonstrate that in the absence of Parp1, there were no significant changes observed in PR-SET7 RNA or protein levels, respectively. This finding supports the conclusion that Parp1 is not directly involved in the regulation of PR-SET7 in *Drosophila* contrasting with the findings of Estève et al.'s study (PMID: 36434141). This discrepancy may arise from differing relationships between PARP-1 and PR-SET7, which could cooperate in the context of *Drosophila* development while playing antagonistic roles in specific cell lines or under particular conditions.

We have updated the Results section to explicitly mention this observation:

"Interestingly, in the absence of PARP-1, neither PR-SET7 RNA nor protein levels were affected (Supplemental Fig.S4-5), indicating that PARP-1 is not directly implicated in the regulation of pr-set7. This finding contrasts with recent evidence demonstrating PARP1-induced degradation of PR-SET7/SET8 in human cells (16)."

Furthermore, we have modified the discussion section to address this discrepancy:

"A recent study demonstrated that in human cells overexpressing PARP-1, PR-SET7/SET8 is degraded, whereas depletion of PARP-1 leads to an increase in PR-SET7/SET8 levels (16). However, in our study involving parp-1 mutant in *Drosophila* third-instar larvae revealed a nuanced scenario: we detected a minor but not significant reduction in both PR-SET7 RNA and protein levels (Supplemental Fig.S4 and S5). This outcome stands in stark contrast to the previous study's findings. The discrepancy could be due to the distinct experimental approaches used: the previous research focused on mammalian cells and in vitro experiments, whereas our study examined the functions of PARP-1 in whole Drosophila third-instar larvae during development. Consequently, while PARP-1 may cooperate with PR-SET7 in the context of *Drosophila* development, it could exhibit antagonistic roles against PR-SET7 in specific cell lines and under certain biological or developmental conditions."

We believe that these modifications effectively address the raised concern and provide a more comprehensive understanding of the relationship between PARP1 and PR-SET7 in our study. We hope these clarifications enhance the overall robustness and clarity of our findings.

**Reviewer #2 (Public Review):**
Summary:This study from Bamgbose et al. identifies a new and important interaction between H4K20me and Parp1 that regulates inducible genes during development and heat stress. The authors present convincing experiments that form a mostly complete manuscript that significantly contributes to our understanding of how Parp1 associates with target genes to regulate their expression.Strengths:The authors present 3 compelling experiments to support the interaction between Parp1 and H4K20me, including:(1) PR-Set7 mutants remove all K4K20me and phenocopy Parp mutant developmental arrest and defective heat shock protein induction.(2) PR-Set7 mutants have dramatically reduced Parp1 association with chromatin and reduced poly-ADP ribosylation.(3) Parp1 directly binds H4K20me in vitro.Weaknesses:(1) The RNAseq analysis of Parp1/PR-Set7 mutants is reasonable, but there is a caveat to the author's conclusion (Line 251): "our results indicate H4K20me1 may be required for PARP-1 binding to preferentially repress metabolic genes and activate genes involved in neuron development at co-enriched genes." An alternative possibility is that many of the gene expression changes are indirect consequences of altered development induced by Parp1 or PR-Set7 mutants. For example, Parp1 could activate a transcription factor that represses metabolic genes. The authors counter this model by stating that Parp1 directly binds to "repressed" metabolic genes. While this argument supports their model, it does not rule out the competing indirect transcription factor model. Therefore, they should still mention the competing model as a possibility.

We appreciate Reviewer 2's insightful comments during both rounds of revision, which have significantly enriched the quality of our manuscript. The binding of PARP1 to loci encoding metabolic genes indeed suggests a direct role of PARP1 in their regulation. However, we acknowledge Reviewer 2's point that some of these targets might be regulated indirectly, with PARP1 potentially modulating the expression of intermediary transcription factors.

To address this possibility, we have revised the discussion section of our manuscript accordingly:

"Remarkably, our observations indicate a notable affinity of PARP-1 for binding to the gene bodies of these metabolic genes (34), suggesting a direct involvement of PARP1 in their regulation. Nonetheless, it remains plausible that certain genes may be indirectly regulated by PARP1 through intermediary transcription factors."

We trust that this modification adequately addresses Reviewer 2's concern.

(2) The section on inducibility of heat shock genes is interesting but missing an important control that might significantly alter the author's conclusions. Hsp23 and Hsp83 (group B genes) are transcribed without heat shock, which likely explains why they have H4K20me without heat shock. The authors made the reasonable hypothesis that this H4K20me would recruit Parp-1 upon heat shock (line 270). However, they observed a decrease of H4K20me upon heat shock, which led them to conclude that "H4K20me may not be necessary for Parp1 binding/activation" (line 275). However, their RNA expression data (Fig4A) argues that both Parp1 and H40K20me are important for activation. An alternative possibility is that group B genes indeed recruit Parp1 (through H4K20me) upon heat shock, but then Parp1 promotes H3/H4 dissociation from group B genes. If Parp1 depletes H4, it will also deplete H4K20me1. To address this possibility, the authors should also do a ChIP for total H4 and plot both the raw signal of H4K20me1 and total H4 as well as the ratio of these signals. The authors could also note that Group A genes may similarly recruit Parp1 and deplete H3/H4 but with different kinetics than Group B genes because their basal state lacks H4K20me/Parp1. To test this possibility, the authors could measure Parp association, H4K20methylation, and H4 depletion at more time points after heat shock at both classes of genes.

We sincerely appreciate Reviewer 2 for their insightful comment on our manuscript. Your hypothesis regarding the potential induction of H3/H4 dissociation from group B genes by PARP-1, leading to a reduction in H4K20me1, offers a thought-provoking perspective. However, our findings suggest an alternative interpretation.

Our data indicate that while H4K20me1 is indeed present under normal conditions at group B genes, its reduction following heat shock does not seem to impede PARP-1's role in transcriptional activation (Fig. 4A, C, and E). Instead, we propose that this decrease in H4K20me1 might signify a regulatory shift in chromatin structure, facilitating transcriptional activation during heat shock, with PARP-1 playing an independent facilitating role. Moreover, existing studies have highlighted the dual role of H4K20me1, acting as a promoter of transcription elongation in certain contexts and as a repressor in others.

The elevated enrichment of H4K20me1 in group B genes under normal conditions may indeed indicate a repressive state that requires alleviation for transcriptional activation. Additionally, we cannot discount the possibility of unique regulatory functions associated with PR-SET7, extending beyond its recognized role as a histone methylase. Non-catalytic activities and potential interactions with non-histone substrates might contribute to the nuanced control exerted by PR-SET7 on group B genes during heat stress.

Furthermore, our exploration of *pr-set720* and *ParpC03256* mutants reveals distinct roles for PARP-1 and H4K20me1 in modulating gene expression (Fig 3E). This reinforces the notion that the interplay between PR-SET7 and PARP-1 involves a multifaceted regulatory mechanism.

To address these points, we have revised the discussion section of our manuscript accordingly:

"Another plausible explanation could be that the recruitment of PARP-1 to group B genes loci promotes H4 dissociation and then leads to a reduction of H4K20me1. However, our findings suggest an alternative interpretation: the decrease in H4K20me1 at group B genes during heat shock does not seem to impede PARP-1's role in transcriptional activation, (Fig.4A, C and E). Rather than disrupting PARP-1 function, we propose that this reduction in H4K20me1 may signify a regulatory shift in chromatin structure, priming these genes for transcriptional activation during heat shock, with PARP-1 playing an independent facilitating role. Moreover, existing studies have highlighted the dual role of H4K20me1, acting as a promoter of transcription elongation in certain contexts and as a repressor in others (13, 26, 39, 40, 42-46). The elevated enrichment of H4K20me1 in group B genes under normal conditions may indicate a repressive state that requires alleviation for transcriptional activation. Additionally, we cannot discount the possibility of unique regulatory functions associated with PR-SET7, extending beyond its recognized role as a histone methylase. Non-catalytic activities and potential interactions with non-histone substrates might contribute to the nuanced control exerted by PR-SET7 on group B genes during heat stress (47, 48). Furthermore, our exploration of *pr-set720* and *parp-1C03256* mutants reveals distinct roles for PARP-1 and H4K20me1 in modulating gene expression (Fig 3E). This reinforces the notion that the interplay between PR-SET7 and PARP-1 involves a multifaceted regulatory mechanism. Understanding the intricate relationship between these molecular players is crucial for elucidating the complexities of gene expression modulation under heat stress conditions."

We believe that this modification enhances the clarity of our conclusions and adequately addresses Reviewer 2's concerns regarding the intricate relationship between PARP-1, H4K20me1, and PR-SET7 in transcriptional regulation under heat stress conditions.